# Risk and Pattern Analysis of Pakistani Crime Data Using Unsupervised Learning Techniques

**Faria Ferooz** [1,2,*], **Malik Tahir Hassan** [1], **Sajid Mahmood** [3], **Hira Asim** [1], **Muhammad Idrees** [4,*], **Muhammad Assam** [5], **Abdullah Mohamed** [6] and **El-Awady Attia** [7,8]

[1] Department of Software Engineering, University of Management and Technology, Lahore 54770, Pakistan; tahir.hassan@umt.edu.pk (M.T.H.); hira.asim@umt.edu.pk (H.A.)

[2] Department of Legal Studies, Alma Mater Studiorum—Università di Bologna, 40126 Bologna, Italy

[3] Department of Informatics and Systems, University of Management and Technology, Lahore 54770, Pakistan; sajid.mahmood@umt.edu.pk

[4] Department of Computer Science and Engineering, University of Engineering and Technology Lahore, Narowal Campus, Lahore 54400, Pakistan

[5] College of Computer Science and Technology, Zhejiang University, Hangzhou 310027, China; assam@zju.edu.cn

[6] Research Centre, Future University in Egypt, New Cairo 11835, Egypt; mohamed.a@fue.edu.eg

[7] Department of Industrial Engineering, Prince Sattam Bin Abdulaziz University, Al Kharj 16273, Saudi Arabia; e.attia@psau.edu.sa

[8] Mechanical Engineering Department, Faulty of Engineering (Shoubra), Benha University, Cairo 13511, Egypt

* Correspondence: faria.ferooz@umt.edu.pk (F.F.); midrees10@uet.edu.pk (M.I.)

**Abstract:** To reduce crime rates, there is a need to understand and analyse emerging patterns of criminal activities. This study examines the occurrence patterns of crimes using the crime dataset of Lahore, a metropolitan city in Pakistan. The main aim is to facilitate crime investigation and future risk analysis using visualization and unsupervised data mining techniques including clustering and association rule mining. The visualization of data helps to uncover trends present in the crime dataset. The K-modes clustering algorithm is used to perform the exploratory analysis and risk identification of similar criminal activities that can happen in a particular location. The Apriori algorithm is applied to mine frequent patterns of criminal activities that can happen on a particular day, time, and location in the future. The data were acquired from paper-based records of three police stations in the Urdu language. The data were then translated into English and digitized for automatic analysis. The result helped identify similar crime-related activities that can happen in a particular location, the risk of potential criminal activities occurring on a specific day, time, and place in the future, and frequent crime patterns of different crime types. The proposed work can help the police department to detect crime events and situations and reduce crime incidents in the early stages by providing insights into criminal activity patterns.

**Keywords:** crime analytics; clustering; risk identification; frequent pattern mining; data mining; public safety

## 1. Introduction

Crime is an important social issue that affects a society's quality of life and economic growth in a negative manner. Criminal activities are on the rise globally, in general, putting the safety and security of the people at stake [1]. In this era of technology, computerised systems are commonly used to monitor criminal activities [2]. Both criminologists and laypeople are becoming increasingly concerned about crime hotspots. While crime prevention theory and research have previously focused on the larger worlds of the community and the neighbourhood, there is an increasing recognition of changing that to concentrate solely on the small worlds in which the characteristics of an area and its everyday activities combine to create crime events [3].

Crimes can be of many types such as (1) 'Crime against the person' which includes murder, attempt to murder, hurt, assault on a public servant, kidnapping, etc.;(2) 'Crime against belongings' consists of crimes like robbery, phone snatching, etc;(3) 'White-collar crimes' are non-violent crimes which are committed for financial gains such as fraud; and (4) 'victim-less crimes' which include drug addiction, suicide, etc. The law enforcement agencies provide security to the public to prevent incoming public threats and to maintain the law and order situation in the country [4].

The law enforcement agencies at the provincial level consist of police teams from different states. Likewise, in Pakistan, police teams of Punjab, Sindh, Baluchistan, Khyber Pakhtunkhwa (KPK), and Gilgit Baltistan have constituted the law enforcement agencies at the provincial level, while at the federal level, the law enforcement agencies include the Federal Investigation Agency (FIA), Anti-Narcotics Force (ANF), National Counter Terrorism Authority (NCTA), and National highway and motorway police (List of law enforcement agencies in Pakistan: https://bit.ly/3lrwmVM (accessed on 28 October 2020), etc.

The Pakistan Bureau of Statistics (Pakistan Bureau of Statistics official webpage: http://www.pbs.gov.pk/ (accessed on 30 October 2020) has maintained records of criminal activities from 2012 to 2017 in different states of Pakistan. The Crime Bureau of Statistics updates the crime figures according to the crime type and occurrence ratio in every province/state.

Table 1 shows the statistics of the crime types reported in different provinces/states of Pakistan in 2017 (Crimes reported by the Pakistan Bureau of Statistics: https://bit.ly/3fqgZJq (accessed on 30 October 2020). As per the stated information, it can be seen that the overall crimes reported in Pakistan in the year 2017 are 683,925 in number.

**Table 1.** Crimes reported by the Pakistan Bureau of Statistics in the year 2017 (Crimes reported by the Pakistan Bureau of Statistics: https://bit.ly/3fqgZJq (accessed on 30 October 2020).

| | Provinces/States | | | | | | | | Overall, in Pakistan |
|---|---|---|---|---|---|---|---|---|---|
| | **Punjab** | **Sindh** | **KPK** | **Baluchistan** | **Islamabad** | **Railways** | **G. B** | **AJK** | |
| **Murder** | 3914 | 1409 | 2361 | 325 | 84 | 6 | 61 | 75 | 8235 |
| Attempt to Murder | 4440 | 1644 | 2641 | 333 | 163 | 5 | 94 | 179 | 9499 |
| Kidnapping/ Abduction | 13,558 | 2927 | 1197 | 248 | 99 | 7 | 48 | 279 | 18,363 |
| Dacoity | 602 | 572 | 45 | 38 | 16 | 1 | 4 | 2 | 1280 |
| Robbery | 9385 | 2364 | 276 | 185 | 195 | 4 | 15 | 34 | 12,458 |
| Burglary | 11,023 | 1344 | 798 | 135 | 253 | 1 | 63 | 216 | 13,833 |
| Cattle Theft | 4721 | 383 | 126 | 39 | 22 | 0 | 17 | 34 | 5342 |
| Other Theft | 33,053 | 2221 | 882 | 272 | 506 | 234 | 64 | 72 | 37,304 |
| Others | 325,149 | 57,409 | 172,504 | 7917 | 5798 | 1270 | 1370 | 6194 | 577,611 |
| Total Recorded Crimes | **405,845** | **70,273** | **180,830** | **9492** | **7136** | **1528** | **1736** | **7085** | **683,925** |

According to the findings of the Crime Bureau of Statistics over the past six years, it has been analysed that the ratio of crimes like murder, kidnapping, robbery, fraud, sex abuse, gang rape, etc., has increased tremendously. The Pakistan Bureau of Statistics has stated that the total crimes reported in 2014, 2015, 2016, and 2017 in each province are 627127, 633299, 677554, and 683925 in numbers, respectively (Crimes reported by the Pakistan Bureau of Statistics: https://bit.ly/3fqgZJq (accessed on 30 October 2020)).

According to the Pakistan Bureau of Statistics report, it can be seen that the crime rate in Pakistan is increasing at a very high pace. The proposed work is an initiative to help the concerned agencies in the detection and prevention of criminal activities using exploratory data mining techniques.

The availability of crime-related data is limited. Thus, in this research, actual crime-related data are gathered from different police stations in Lahore, Pakistan with the intent of

reducing the crime rate in the concerned areas. In this study, the First Information Reports (FIR) data are initially collected in a complex format written in the Urdu language. The data of the FIRs are gathered from three different police stations in 'Mustafa town', 'Johar Town', and 'Mughalpura' areas of Lahore, Pakistan. The collected data of the FIRs are converted into the English language for attribute creation and feature selection. After completing the data pre-processing phase, the structured dataset has consisted of 24 attributes in total. The attribute selection is performed on the dataset to choose only those attributes that can help perform the cluster analysis and frequent pattern mining. The finalised dataset taken into consideration for the exploratory analysis has 636 records and six attributes in total [2017–2018]. Data visualisation analysis is applied to the finalised crime dataset to analyse the overall structure and trends present in the dataset. After getting insights from the data, data mining techniques named cluster analysis and association rule mining are applied to the crime dataset for exploratory analysis.

In this research, we have proposed an interdisciplinary strategy between computer science and criminal justice to discover crime patterns using data mining techniques that can assist the concerned officers in fixing and solving offences more quickly. Data mining methods have flourished to predict the cyber-crimes in banking sectors [5], threat analysis and prediction [6], and cyber-crime analysis in social media [7]. For the identification of similar crime characteristics and risky areas, the k-modes clustering algorithm is applied. To find frequent crime patterns related to crime type, crime date, and time, association rules mining algorithm named Apriori algorithm [8] is applied to the generated finalised dataset.

The contributions of the paper are summarized as follows:

1. Crime data set is gathered from three police stations in Lahore, Pakistan, in the form of FIRs written in the Urdu Language, as shown in Figure 1. The collected data are then translated into an English language format to perform the intended analysis;
2. The data were initially presented in an unstructured format, so the information is then converted into the tabular format as shown in Table 2 to perform data pre-processing and visualisation;
3. Different data pre-processing techniques are applied for data cleaning and extraction of task-relevant attributes presented in structured data;
4. Data visualisation analysis is performed to find the trends present in the dataset;
5. Finally, unsupervised learning techniques are applied to the dataset. Cluster analysis is performed to identify similar crime characteristics and risky crime locations. Moreover, the association rule mining technique extracts the frequent patterns to analyse criminal activity.

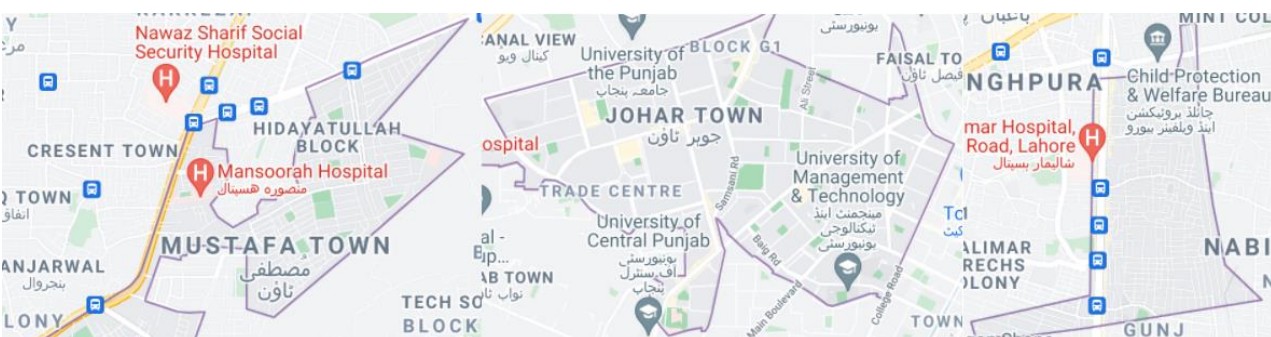

**Figure 1.** Map of the study area of three different towns.

The primary motivation behind performing the exploratory analysis on the crime dataset is to find out the hidden trends and practices of the criminal activities, which can help the police authorities of Pakistan reduce and prevent criminal actions. The solutions proposed in this research work can help make Pakistan a crime-free, safe, and peaceful nation.

This paper is categorized into the following sections. Section 2 describes the literature review. Methodology and experimental analysis are discussed in Section 3. Results and discussion are presented in Section 4. Limitations are discussed in Section 5. Finally, Section 6 narrates the conclusion and future research agenda.

**Table 2.** Sample attributes names along with values of the generated structured dataset.

| Form No. | Serial-No. | Town | District | Event Happening Date | Event Happening Time | Crime-Reporting Date | Crime-Reporting Time | Crime Type |
|---|---|---|---|---|---|---|---|---|
| from-5–24 | 665 | Johar town | Lahore | 10 February 2018 | 2:00 p.m. | 16 February 2018 | 1:40 p.m. | Theft |
| from-5–24 | 797 | Mustafa town | Lahore | 15 April 2018 | 1:10 p.m. | 16 April 2018 | 11:40 a.m. | Theft |
| from-5–24 | 788 | Mughalpura | Lahore | 3 September 2018 | | 14 April 2018 | 12:45 p.m. | Robbery |
| from-5–24 | 779 | Mustafa town | Lahore | 4 September 2018 | 10:30 a.m. | 4 November 2018 | 4:15 p.m. | Fraud |
| from-5–24 | 663 | Mughalpura | Lahore | 2 May 2018 | 7:00 a.m. | 15 February 2018 | 9:10 p.m. | Assault |
| from-5–24 | 664 | Johar town | Lahore | 15 February 2018 | 11:55 p.m. | 16 February 2018 | 12:05 a.m. | Theft |
| from-5–24 | 668 | Mustafa town | Lahore | 15 February 2018 | | 16 February 2018 | 2:30 p.m. | Robbery |

## 2. Literature Review

This section discusses recent research articles related to the data mining techniques applied to the crime datasets. Moreover, the limitations of the studies are also highlighted.

In [9] authors have used the open data set of san Francisco from 2003 to 2015. The supervised learning algorithms Decision tree, k-nearest neighbour (KNN), and random forest are used to predict crime type. Even though the Decision tree and k-nearest neighbour (KNN) did not perform well on the data set, the random forest algorithm outperformed the dataset and the accuracy level achieved was 99.16%. However, this work has focused on the predefined data crime set. Grouping of similar crime event locations and the identification of frequent crime patterns is the limitation of this research addressed in our paper.

The methodology applied in [10] has analysed murder crimes reported in India in 2013 and 2014. Murder crime-related data were extracted from the official websites of the NDTV, NCBR, and CNN-IBN news. The motive of the research was to find the alarming locations where the murder crime rate is high. Topic Modelling and Named Entity Recognition analysis were applied to the dataset. Topic Modelling was applied for the identification of latent text patterns from the content. The solutions provided by the author were proven very helpful for analysts to find the alarming locations. However, the work has been done on the predefined crime data set and the identification of similar crime characteristics is still un-discovered in this paper, which is one of the significant contributions of our work.

The authors in [11] have applied the theoretical model based on data mining techniques on the crime dataset of England from 1990 to 2011. Data mining techniques are applied to analyse, investigate, and discover crime patterns of different crime types. The clustering algorithm is used to group similar crime patterns. Various classification algorithms are applied to predict the crimes that are expected to happen soon. A genetic algorithm is applied for the optimisation of the results. However, a grouping of similar crime locations and frequent crime pattern identification is the limitation of this research work to which our paper contributes.

Multiple data mining techniques have been applied in [12] to Indian crime data for crime detection. The unstructured data were collected from various web sources from 2000 to 2012 and then processed afterward. Clustering and classification algorithms are applied to perform the analysis. To group similar criminal activities, clustering analysis using the k-means algorithm was performed. The classification algorithm K-Nearest Neighbour was applied to predict crimes that were expected to happen in the future. The accuracy reported in the paper is 93.2%. This paper focused on the pre-defined dataset, whereas we have generated our dataset from FIRs that were written in the Urdu language, and frequent crime event patterns are identified in our work.

An effort has been made to detect the crime patterns by applying data mining techniques on Chicago and NSW crime datasets from 2008 to 2012 and from 2013 to 2017 [13].

The data are pre-processed, and Fuzzy Association rule mining is applied to the dataset to generate crime patterns. The between among the criminal activities of different cities is identified and analysed. In our research, we have also identified similar crime characteristics and potential dangerous crime locations.

The primary purpose of the research [14] was to explore different techniques for detecting crime patterns, which potentially can help the police department solve crime cases. In this research, crime patterns are extracted from data using a clustering algorithm. K-means clustering algorithm is applied to the dataset to group similar types of crime characteristics and criminal features. The crime types are grouped with similar suspect and victim features like victim age, suspect age, etc. The solutions provided by the author proved very helpful for analysts to solve crime cases by analysing the crime patterns. However, the identification of risky locations is still undiscovered in this paper, which is one of the significant contributions of our work.

In [15], it has been reported that the Law Enforcement Agency of the United States has collected crime information using well-known resources like the FBI's CRS (Uniform Crime Reporting System) and its successor, the National Incident-Based. The collected records contain 1.3 million suspects and criminal records from 1970 to 2004 from the Tucson police department. Data mining techniques are applied to the crime dataset to make predictions using clustering and frequent patterns analysis. The relation between different criminal groups is identified using extracted data from incident summaries. Clustering algorithms are applied to identify criminal subgroups. The block modelling approach and association rule mining are applied to determine extracted subgroups' interaction patterns and rules. However, the patterns and grouping of criminal activities expected to happen particular location is the limitation of this work covered in our contribution.

In [16], the authors aimed to reduce the dimensionality of the datasets using feature selection algorithms in performing association rule mining. To do so, Relief-Based algorithms (RBAs) are used for the feature selection process. This study has proposed a model that tends to highlight the features positively correlated with the class attributes for performing association rule mining activity. Multiple experiments conducted on real-time datasets have reported that the designed model has shortlisted a concise and related subset of features as compared to the other models presented in the literature study. The results shown in the paper have also extracted the factors that can be proven helpful toward students' academic growth and success in higher education. However, in our work, we applied association rule mining to the crime dataset to observe its performance.

A machine learning algorithm has been applied to the crime data of Dallas, a city in Texas [17]. The data are available on Dallas's open data portal. The data have been split into training and testing sets to perform classification. The long-term crime forecast for robbery crime is generated at 200 by 200 feet grid cells. Random Forest and Kernel density estimation classification techniques outperformed in terms of forecasting future crimes. The results reported in the paper are promising and can also be used to predict future crimes. However, identifying the other factors involved during the crime execution is the limitation that our research has addressed.

Predictions have been made to depict the crime rate by applying the decision tree classifier to the crime dataset [18]. The primary purpose of the research was to identify the crime characteristics and the level (high, medium, and low) of crime. J48 decision tree classification algorithm was applied to predict future crimes and the likelihood of the country counted in a low, medium, and high category related to crime. The accuracy of the J48 algorithm reported in the paper is 94% which can be considered reliable to predict future crimes. However, patterns of criminal activities are unaddressed by the form which is addressed in our contribution.

Recent research conducted in [1] has extracted the crime data of the past eight years from web news archives to perform predictive analysis on the criminals' network behaviour. The results of the study were further used as an input to perform classification. The K-Nearest Neighbour and Random Forest classifiers achieved accuracies of 92% and 64%,

respectively. The overall research has highlighted that robbery is the most prevalent occurring criminal activity in Pakistan. This research lacks the factor of identifying similar criminal activities, risky crime locations, and frequent crime patterns, which are addressed by our study.

Another contribution toward serial killing criminal activity is made in [19]. The authors tried to link the criminal actions committed by an offender with the help of a decision support system. The proposed system works on pairwise classification. Feature similarity algorithms are developed to compute pairwise similarities to determine the serial killing crime cases. The developed method is tested on the real-world crime data of robberies and showed promising results. Currently, the system is working publicly in the Security Bureau of China. Serial killing crimes are the only focus of this research, whereas we have targeted all possible criminal activities that can be expected to happen in a particular location at a specific time.

Zhuan et al. [20] in the United States have conducted research to predict crime hot spots and patterns of criminal activities with the help of embedded spatial information by a Spatio-Temporal Neural Network (STNN). The data used in this study were provided by the Portland Oregon Police Bureau (PPB) from the past five years. The authors' primary focus was only on identifying patterns and crime locations, whereas, in our work, we have also identified similar criminal activities that are expected to happen in hot crime locations.

In [21], six classification algorithms are applied to crime data using the WEKA tool to classify different crimes based on the population. The data are gathered from the Federal Bureau of Investigation of Pakistan. The classification results have indicated that the theft and property crime rate is high in populated areas among all crime types. The main aim of this study is to help law enforcement agencies in predicting and analysing criminal activities Weak Ahead Electricity Power and Price Forecasting using Improved DenseNet-121 Method n addition, our objective is to reduce criminal activities before they occur by analysing crime patterns and risky crime locations.

Research in [22] has been conducted on two UK neighbourhoods to address organised crimes. The police record the effect of organised crime to analyse organised crime and its impact on communities. According to the findings, the planned crime groups are accountable for a much more extensive range of harm than uttered by the national police insights mechanism. This paper only focused on the impact of crime, whereas our primary focus is the reduction of crime rates by predicting them before happening in a particular location.

To analyse and map criminal activities in Pakistan, [23] has investigated the role of Geospatial Technologies. Crime data of the Bhakkar district in the year 2017 have been collected from eleven police stations. Different types of Geographic Information System (GIS) analyses are performed on the data using ArcGIS software. Hot Spot analysis is performed to identify better patrolling areas for the police officers. Moreover, violent and risky crime locations are also determined using the spatial and temporal distribution of the crime data. However, patterns of criminal activities are unaddressed by the paper, which is considered in our contribution.

The author in [24] has analysed neighbourhood relations in Kyoto to test their already developed model for preventing social capital and community-based crimes. The theoretical Structural Equation Modelling (SEM) was used to measure the factors for preventing the selected crimes. The survey data and police records for street crimes and residential burglary suggest that specific efforts by community residents can enhance social capital and lead to security and community safety. Social capital has a great impact on minimising the ratio of street crime. However, the analysis indicated that social capital decreases crime anxiety and enhances awareness in families and others.

The study in [25] applied different data mining techniques to the San Francisco crimes dataset. The dataset includes information regarding the date, crime type, location, etc. The crime analysis reported in the paper is performed for San Francisco to help their analyst solve crime cases by analysing crime patterns. Two classification algorithms, K nearest neighbour (KNN) and Naïve Bays were applied to the crime dataset to predict crimes at a

particular time in a specified location. For testing purposes, K fold cross-validation was performed. The paper reports an accuracy of 70%, which is the result of Naïve Bayes and K fold cross-validation. However, a grouping of similar crime event locations and frequent crime pattern identification is the limitation of this research work which is addressed in our paper.

In [26], data on reported crimes and accidents in Denver, USA are collected from 2011 to 2015. The trends and patterns of crime and accident-related activities are detected and highlighted. Multiple classification algorithms are applied to the dataset to get the desired results. The results, TP Rate, FP Rate, Precision, Recall, and F-Measure values were computed. Additionally, year-wise crime reports of the city were analysed to find out the months when maximum crimes and accidents occurred.

The authors in [27] tried to predict the crime rate by monitoring human behaviour, while in the previous research, criminal activities and crime patterns were indicated using historical data. The crime dataset used by the authors contained reported crimes in the United Kingdom. The crime dataset contained information about the month, year, location, crime type, etc. Additionally, another dataset was created, which includes the demographic data of UK citizens. The classification algorithm random forest is applied to the selected features to analyse the crime ratio of specific months. The performance metrics named accuracy, F1-score, and AUC are also reported in the paper. The results presented have attained an accuracy of 70%. However, the work has been done on the predefined crime data set and the identification of the grouping of similar crime characteristics is still undiscovered in this paper, which is one of the important contributions of our work.

The research work presented in [28] extracted and analysed the crime patterns of each criminal individually to help the police officers and analysts in solving the reported crime cases. Crime data from the Cambridge police station were collected and crime patterns were identified by applying the Series-Finder algorithm. Series-Finder algorithm output unique and common patterns of the dataset and analysed the patterns showing the best characteristics. This research is a great initiative towards time and labour power-saving. However, the identification of risky locations is still undiscovered in this paper, which is one of the significant contributions of our work.

The authors in [29] have contributed to analysing the country and individual-level influences of criminal victimisation. To measure criminal victimisation, Crime Victims Survey (ICVS) has been used from 57 countries. Two types of surveys were conducted nationwide to measure criminal victimisation. Previous studies mostly focused on violent crime (i.e., homicide), but in this research street crimes such as robbery and residential crimes such as burglary were selected to identify the differences and similarities between them. The results indicated that a higher level of social capital minimises the likelihood of robbery victimisation and has no important effect on robbery victimisation. This paper only focused on the impact of crime, whereas our primary focus is the reduction of the crime rate by predicting them before happening In a particular location.

Different approaches to data mining are applied in [30] on crime data associated with the sheriff's organisations. The semi-supervised learning techniques are applied to identify the significant attributes of the crime data. Clustering techniques are used to group similar crime types and determine the crime patterns present in the dataset. The analysis made by the authors contributes to increasing the predictive accuracy and helps detectives and concerned police officers to solve crime cases faster. However, the identification of risky crime event locations is still undiscovered in this paper, which is one of the important contributions of our work.

In this research work, the crime dataset is gathered from three different police stations in Lahore, Pakistan. The unstructured data is converted into a structured format. The dataset is further pre-processed, visualised, and analysed through an exploratory study. This research aims to visualize the crime hotspot. The exploratory analysis has been made using un-supervised data mining techniques, i.e., clustering and association rule mining. It is used to extract the group's similar crime characteristics, and risk identification of criminal

activities that can be expected to happen in a particular location and frequent patterns of criminal activities. This paper investigates the detailed analysis of crime-related activities and provides multiple solutions to reduce crime rates and incidents.

## 3. Methodology and Experimental Investigation

The data are gathered from different police stations in Lahore, Pakistan. The unstructured data are converted into a structured format, and data mining pre-processing techniques are applied to the dataset. Secondly, task-relevant attributes are selected in consultation with the domain experts to perform visualisation and exploratory analysis. Thirdly, visualisation analysis is applied to get insight from the data. Finally, exploratory research is performed using the k-modes clustering algorithm and association rule mining's Apriori algorithm to extract the factors and patterns involved in criminal activities. The detail of the whole process is discussed in the following sub-sections.

### 3.1. Data Collection and Preprocessing

For this research, data are gathered by visiting the police stations of three different areas named 'Mustafa Town,' 'Johar Town,' and 'Mughalpura' of Lahore, Pakistan as shown in Figure 1. A total of 636 FIR records were collected from police stations written primarily in the Urdu language on papers, while very few records were also present in mixed form, as shown in Figure 2.

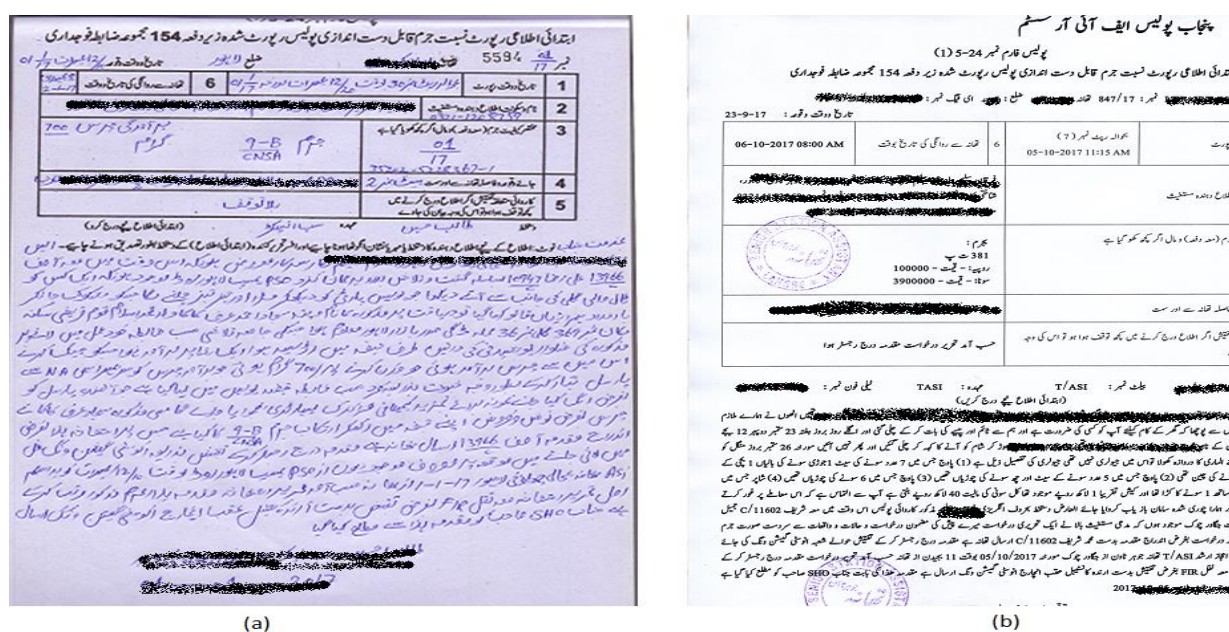

(a)  (b)

**Figure 2.** (**a**) Sample Handwritten FIR Data.(**b**) Sample Computerized FIR Data.

For the identification of the attributes, all collected records are converted into the English language first and then the statements of each record are segregated into multiple attributes to convert data into a structured tabular format. The transformed structured dataset consists of 29 attributes(*form no, Serial-No, Town, District, Event happening date, Event happening time, crime-reporting date, crime reporting time, crime-investigation date, crime-investigation-time, crime-reported-person-details, crime-reported-person-address, CNIC, phone-no, criminal act, crime Type, Item, Model no, Item price, crime description, Crime Event Location, Distance of Crime Event from Police Station, Direction of Crime Event from Police Station, crime-registered-person-name, Belt No, Designation, Phone no, Criminal Name, Address*) in total. The names and values of some sample attributes are shown in Table 2.

### 3.2. Attributes Selection and Extraction for Final Dataset Generation

After translating the data into the English language and converting it into tabular form, attribute selection is performed in consultation with the domain experts to take into account only those attributes that can help perform the exploratory data analysis. Based on each attribute's information, six attributes are selected in total, which became part of the finalised crime dataset.

For better data analysis, data discretisation is done on selected attributes to get more promising results. The attribute *DayType* is derived from the attribute *crime-reportingdate* which has now only two possible values 'Weekdays' and 'Weekend' after discretisation. Similarly, another attribute *Crime Time* is derived from the attribute *crime-reporting-time* that presently has only five possible values, i.e., 'Morning', 'After-Noon, 'Evening', 'Night', and 'Mid-Night'. Table 3 is showing the names and possible values of the selected attributes presented in the finalised crime dataset.

**Table 3.** List of the finalised set of attributes along with possible values.

| Attributes' Name | Attributes' Values |
|---|---|
| *Crime Time* | 'Morning', 'After-Noon', 'Evening', 'Night', 'Mid-Night' |
| *Day Type* | 'Weekdays, Weekend' |
| *Crime Event Location* | 'Abbas Block', 'Azam Garden', 'BOR Main Road', 'Bohar Wala Chowk', 'Canal Bank Rd', 'Canal View Housing Society', 'Ferozpur Road', 'G1 Market', 'Mian Plaza', 'Multan Chungi', 'Samsaani Road', 'Shah Di Khoi', 'Thokar Niaz Baig', 'Township', 'Wahdat Road', 'Sohwari', 'Shalimar Link Road', 'Mughalpura Chowk', 'Mughalpura Road', 'Ramgar Bazar'. |
| *Distance of Crime Event from Police Station* | '1 km', '2 km', '3 km', '4 km', '11 km', '15 km' |
| *The direction of Crime Event from Police Station* | 'North', 'South', 'East', 'West', 'North East', 'South East', 'North West', 'South West' |
| *Crime Type* | 'Assault', 'Conspiracy', 'Drug Trafficking', 'Electricity Theft' 'Fraud', 'Gambling', 'Kidnapping', 'Kite Flying Offense', 'Marriage Ordinance', 'Murder', 'Possession of Weapon', 'Price Control', 'Robbery' 'Theft', 'Traffic Violation' (See Figure 2 for description) |

After performing data discretisation on *Crime Time* and *Day Type* attributes, data characterisation on the attribute *Crime Type* is also done as there exist multiple types of crimes. Figure 3 shows the concept hierarchy of the *Crime Type* attribute.

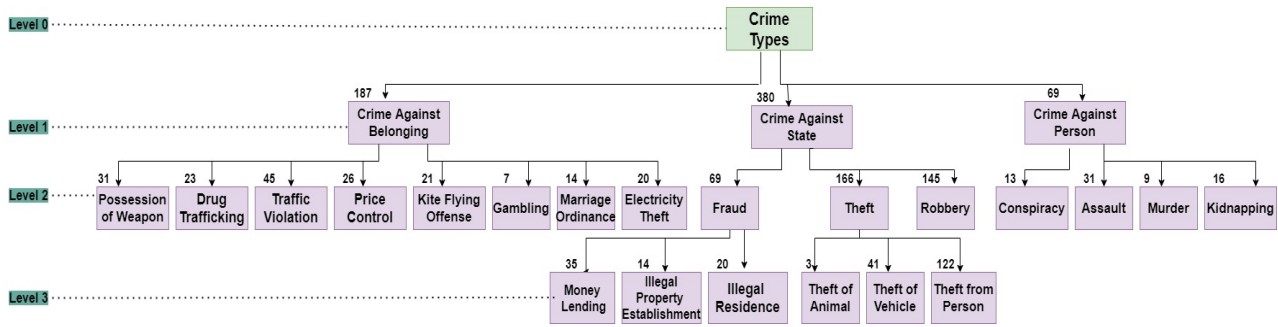

**Figure 3.** Concept hierarchy of attribute Crime Type.

A concept hierarchy is developed to describe the categorization of low-level concepts into high level more generalized concepts. Generalization leads to more pattern extraction. *Crime Type* is initially divided into three groups at Level 1, i.e., "Crime against State",

"Crime against Belongings", and "Crime against Person". Moreover, groups are divided into further subgroups at Level 2 and Level 3, respectively. The numeric value present on each box is showing the count of the records present against that particular *Crime Type*. For example, the crime dataset has 380 records against the *Crime Type* 'Crime Against State'.

### 3.3. Data Visualization Analysis on Selected Attributes

To get insight from the finalised processed crime dataset, data visualisation analysis is performed on selected attributes against the possible values shown in Table 3 above. The *Crime Event Location* is visualized in order to find the highly sensitive areas of crime. The frequency of crime in each location is shown in Figure 4.

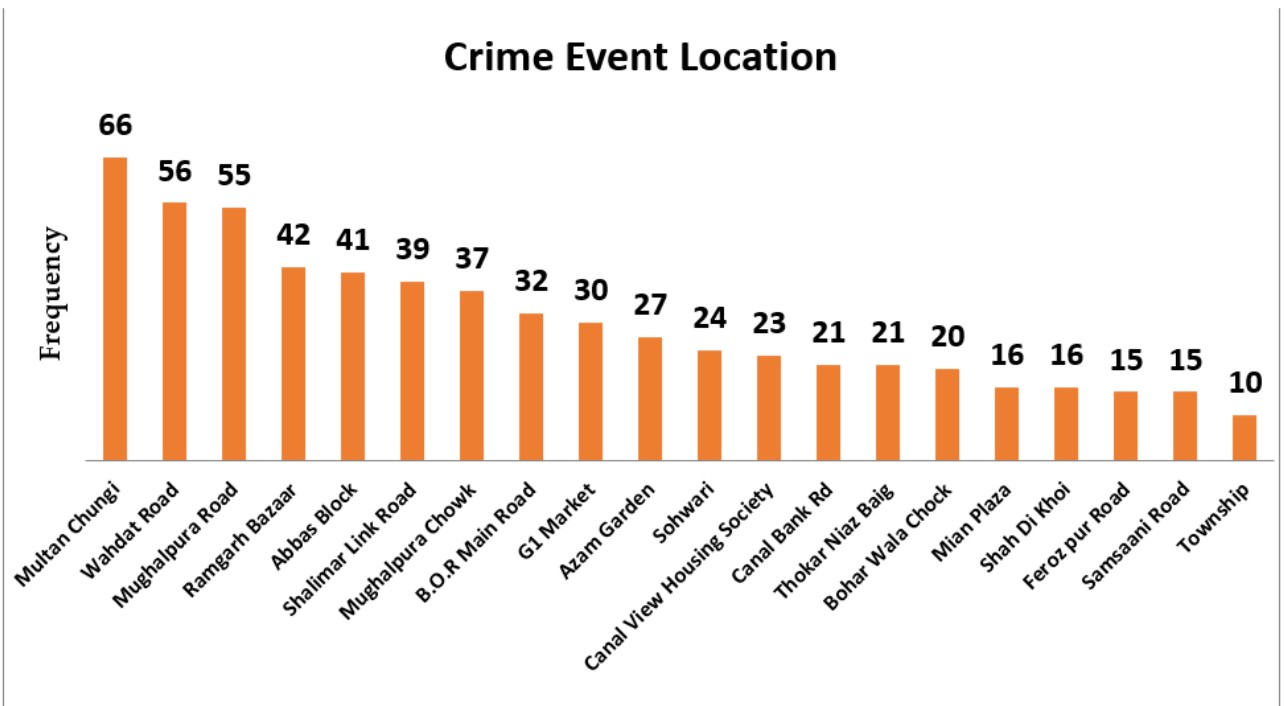

**Figure 4.** Frequency of Crime on each Crime Event Location.

It can be depicted that the location which is targeted most of the time is 'Multan Chungi' and 'Wahdat Road'. However, the ratio of crime rate is low in locations 'Township', 'Samsaani Road', and 'Ferozpur Road'. The visualization of crime event locations is analyzed to help the general public to choose the safer place. Furthermore, the agencies can better plan to enhance the security of these locations.

The attribute *Crime Time* is analysed to find out the occurrence of crime at a particular time as shown in Figure 5.

It can be clearly observed that the rate of crime events that happened in 'Morning', 'Night', and 'After-Noon' is high, compared to 'Evening' and 'Mid-Night'. The overall rate of crime events that happened during the 'Morning' time is high. Most people go to work and children go to school during this particular time, so it could be the reason that the crime ratio is high in 'Morning' time.

Moreover, the trends of the types of criminal activities that occur on a particular crime day, and the visualisation analysis of the attributes of *Crime Day* and *Crime Type* are performed and shown in Figure 6.

As per the visualisation shown in Figure 6, it can be depicted that the 'Robbery' and 'Theft' crime is high on the 'Weekend' because most people go shopping, out for dinners, and visit their families on these days. Thus, criminals mostly plan to execute these crimes on the 'Weekend'. However, the 'Fraud' crime is high on 'weekdays', because mostly

business-related deals get done on 'Weekdays'. These types of factors should be considered, and security measures should be taken accordingly.

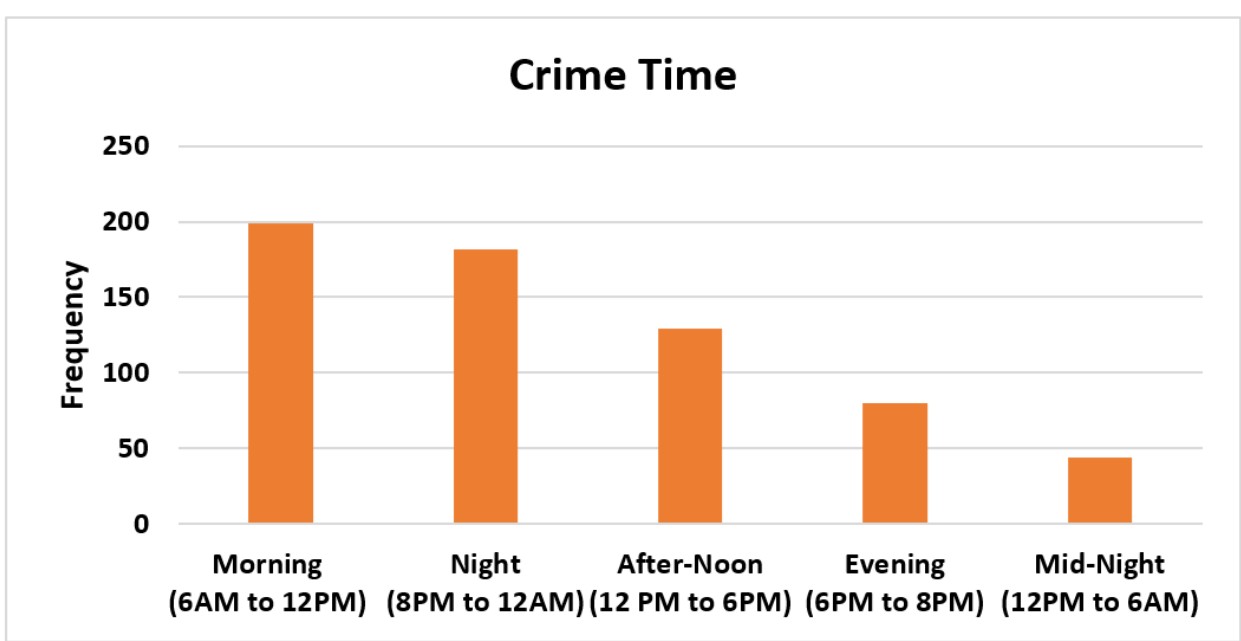

**Figure 5.** Visualisation of occurrence of crime at a particular time.

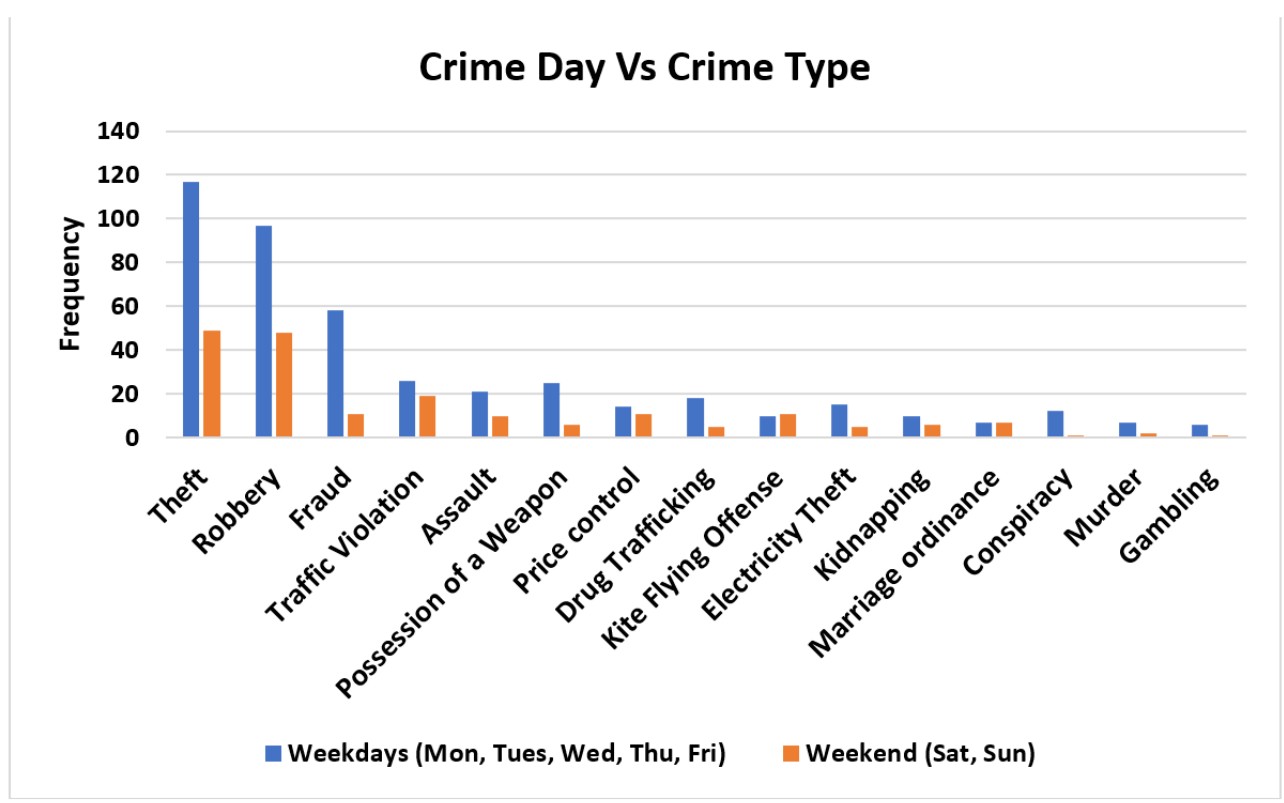

**Figure 6.** Visualization of the distribution of crime types at particular crime day.

Furthermore, the attribute *Crime Event Location* is visualized to find the occurrence of crime on a particular *Crime Day* as shown in Figure 7. It can be observed that the crime event location 'Multan Chungi' and 'Wahdat Road' are mostly targeted on 'Weekdays', whereas 'Abbas Block', 'B.O.R Main Road', and 'Azam Garden' locations are targeted on the 'Weekend'.

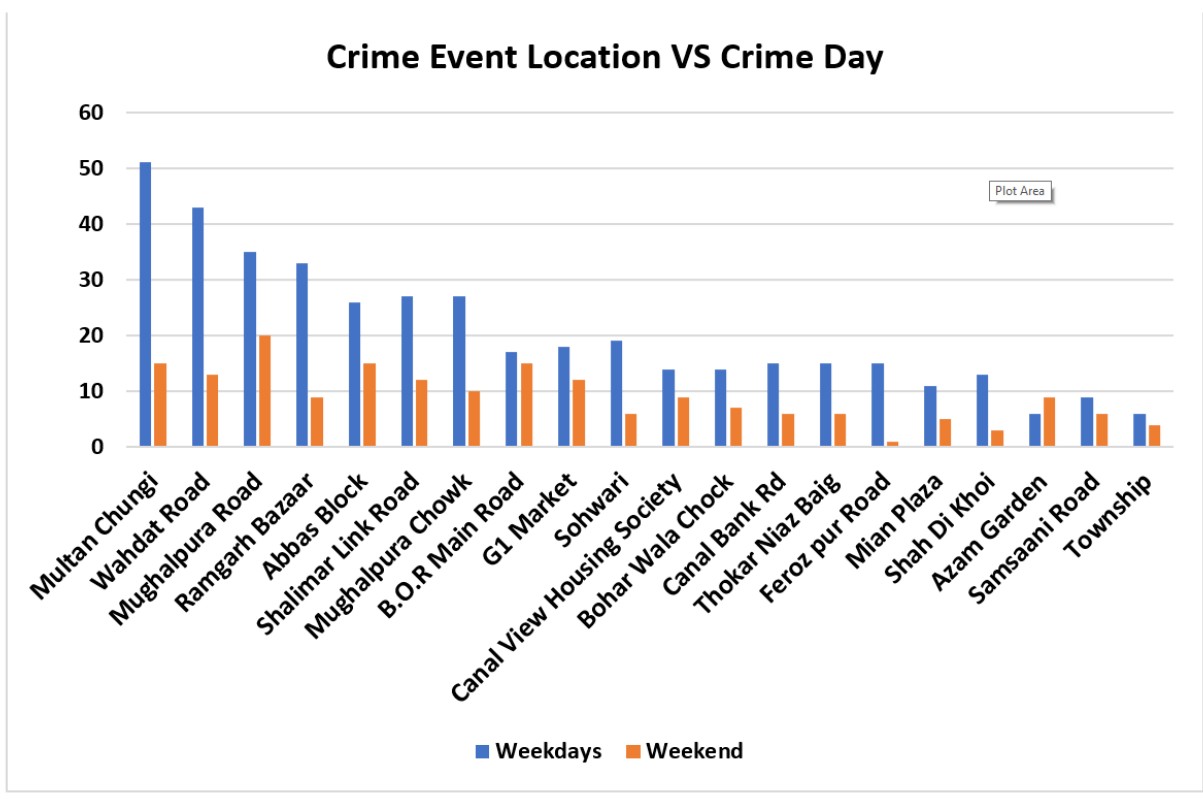

**Figure 7.** Visualization of the distribution of crime event locations on different crime days.

The Visualization analysis is performed on *Crime Event Location* to find out the occurrence of crimes at a different *Crime Time* shown in Figure 8. The ratio of criminal activities is high at location 'Multan Chungi' at 'Night' time, 'Abbas Block' at 'After-Noon', and 'Wahdat Road' at 'Morning' time.

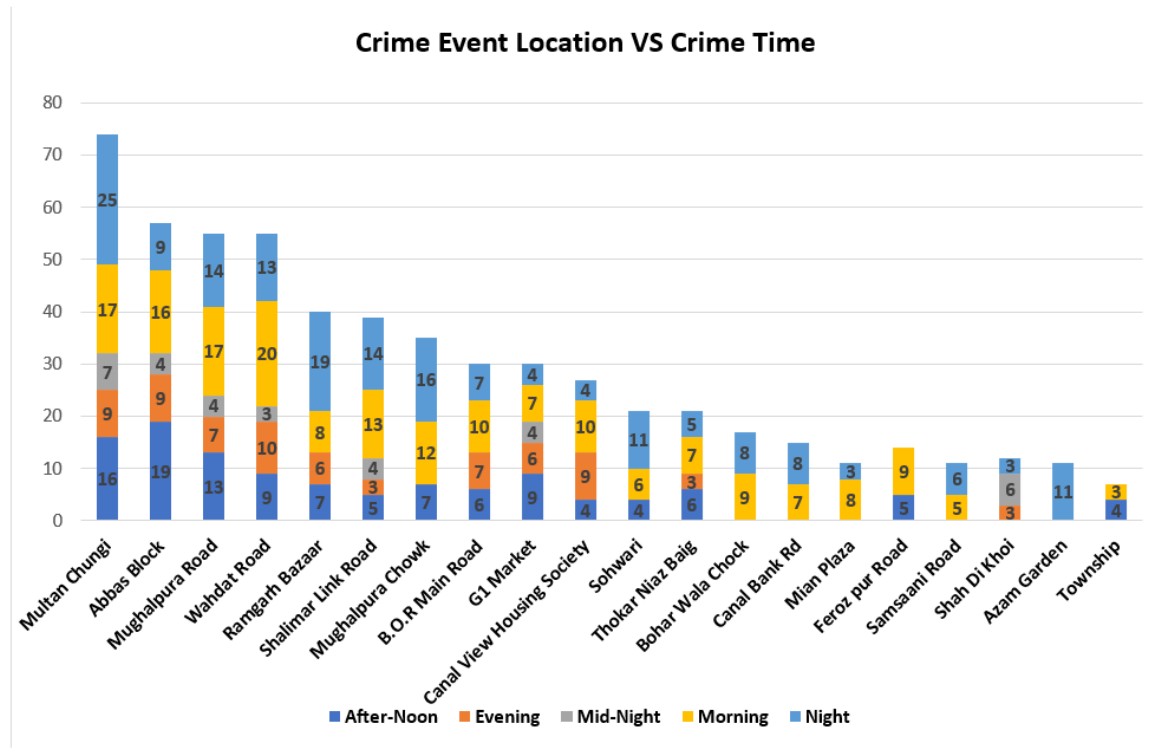

**Figure 8.** Visualisation of Crime Event Location at different crime event times.

The trends of the types of criminal activities that occur at different times of the day, and the visualisation analysis of the attributes *Crime Type* and *Crime Time* is performed and shown in Figure 9.

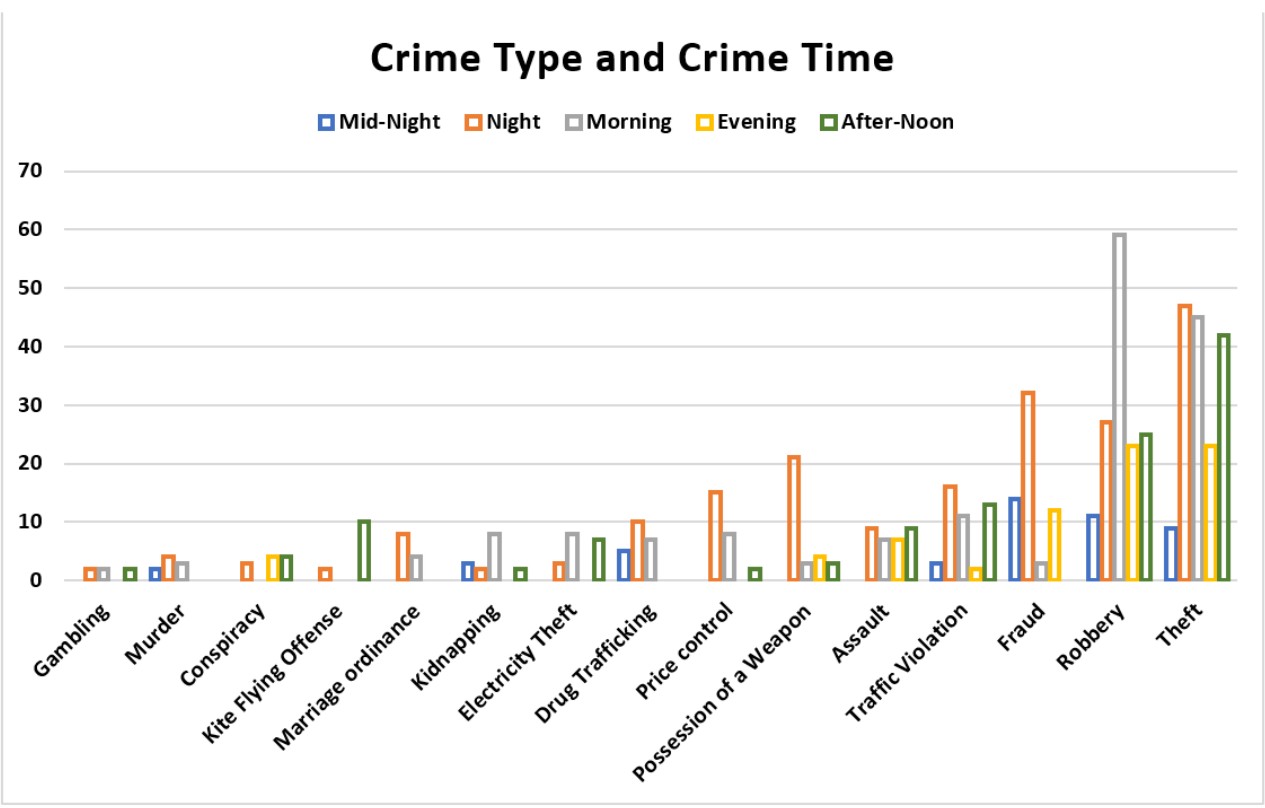

**Figure 9.** Clustered bar chart of the distribution of crime types at different crime event times.

As per the visualisation shown in Figure 9, it can be seen that the 'Robbery' crime rate is high in the 'Morning,' average in the 'Night', 'After-Noon', and 'Evening' time but low at the 'Mid-Night'. However, the 'Theft' crime rate is high in the 'Morning', 'Night', and 'After-Noon' time, average in the 'Evening', but low at the 'Mid-Night' time. Moreover, the 'Fraud' crime rate is also high in the 'Morning' compared to 'Night' and 'After-Noon' time. It can be concluded that the overall crime rate is mostly high in the 'Morning' and 'Night' time.

To find out the targeted areas where specific criminal activities occur mostly, a bar graph has been generated against the attributes *Crime Type* and *Crime Event Location* shown in Figure 10.

It can be seen that the rate of 'Robbery' crime is high on 'Mughalpura Road', and the rate of 'Theft' crime is high on 'Ramghar Bazaar' and the 'G1 Market' location. Moreover, 'Fraud' crime is high in the 'Mughalpura Chowk' area. The frequency of a particular *Crime Type* is also shown in the stacked column chart.

The attributes of *Crime location* and the *Distance of the Crime Event from the Police Station* are visualised and then analysed to find out the trends of the criminal activities and determine whether the criminals target the location near the police station or not. The visualisation analysis is shown in Figure 11.

In the visualisation analysis, trends of criminal activities are identified. Moreover, this analysis can also help the analysts and concerned police officers take necessary security actions on particular locations and events and understand the criminals' crime strategies and targets.

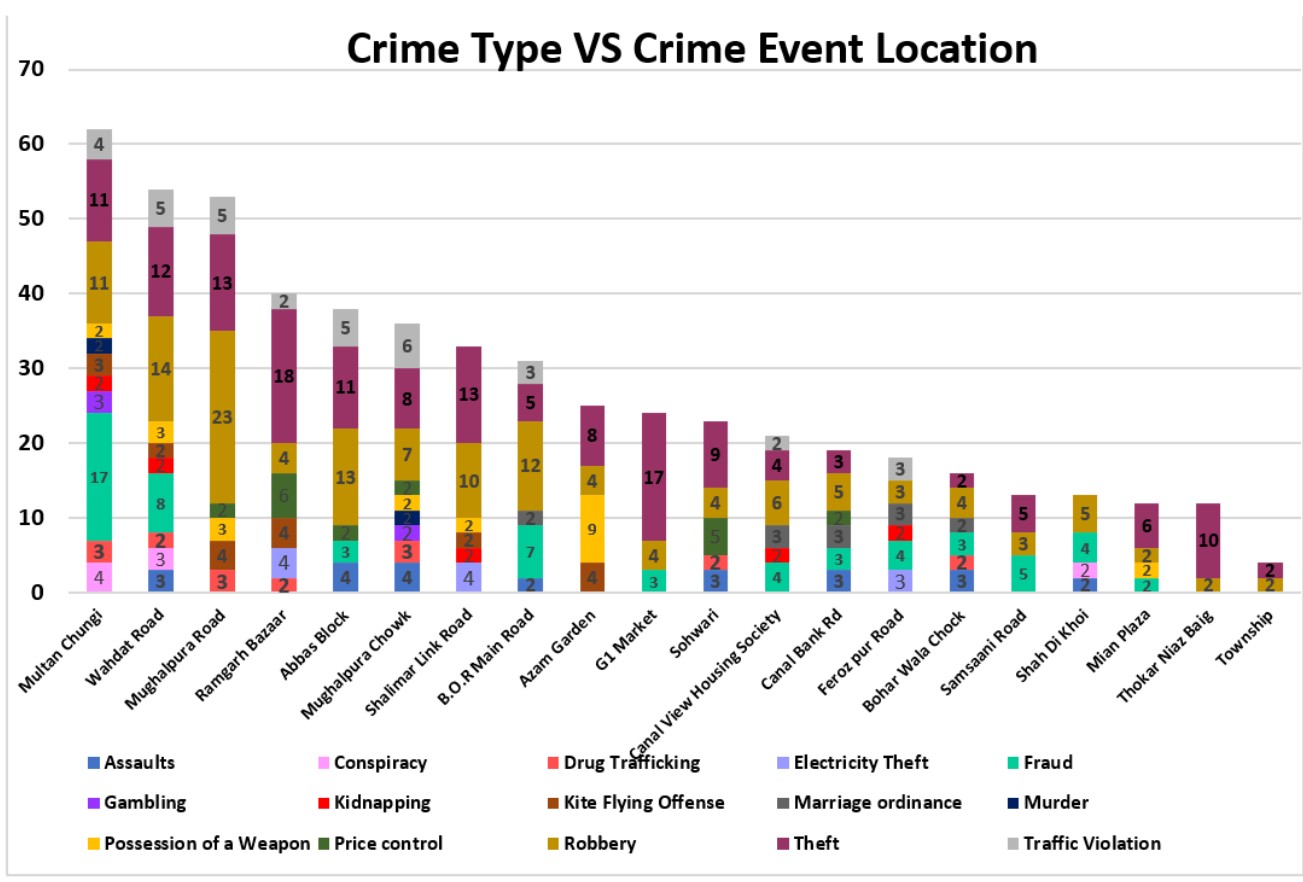

**Figure 10.** Visual analysis of the distribution of crime types at different crime event locations.

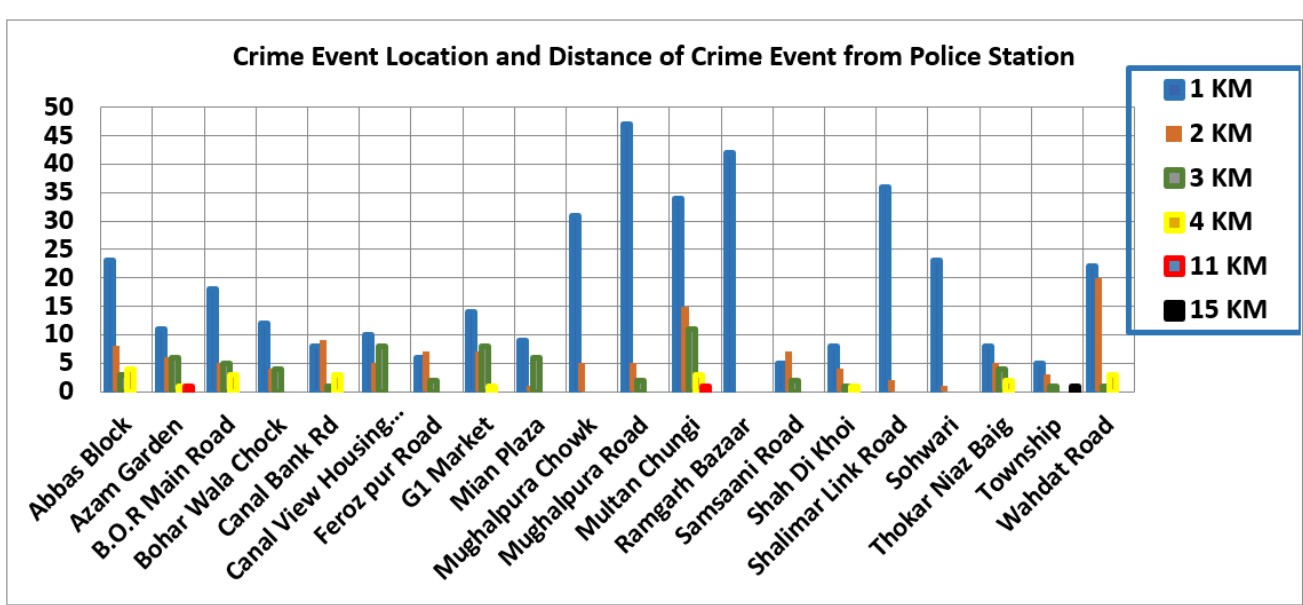

**Figure 11.** Clustered bar chart of the distribution of crime event locations for a distance of crime event from the nearest police station.

### 3.4. Unsupervised Data Analysis

After getting insight from the data using visualisation analysis, cluster analysis and association rule mining are applied to the pre-processed crime dataset.

### 3.4.1. Cluster Analysis

Cluster analysis is the technique used to group similar objects together based on their similar characteristics or how closely they are related. However, in cluster analysis, you do not precisely know how numerous the groups that exist within the data are. The degree of similarity and difference between each object is used to categorise the cluster and assign a cluster group [31].

Crime is a sensitive domain where clustering can play an essential role for analysts to precede and solve the investigation cases faster.The K-modes clustering algorithm is implemented in Python and applied to the dataset to discover similar groups of crimes that occur in a particular location. It can be seen from Table 3 that we have categorical values against the attributes so we are using k-modes in our work. To deal with the categorical data, it employs a simple matching dissimilarity measure, replaces the mean of the clusters with modes, and updates the model in the clustering process using a frequency-based technique to reduce the function cost [32].

The steps of the K Modes algorithm are as follows:

1. Pick up a total of K observations at random;
2. Determine the dissimilarities and allocate each observation to the cluster that is closest to it;
3. Create new modes for the clusters;
4. Repeat steps 2–3 until no re-assignment is required.

Afterward, the obtained clusters are used to identify the risky locations, where criminal activities can potentially happen, to improve safety and security. The details of risk identification and the results of the cluster analysis are explained in the Section 4.

### 3.4.2. Association Rule Mining

Association rule mining is a well-known data mining technique to extract frequent and interesting patterns in a dataset. The Apriori algorithm follows the association rule mining methodology for the generation of frequent item sets [33].Those item sets are considered frequent whose support is higher than the threshold value or specified minimum support. The Apriori algorithm follows the following steps:

1. Define the support for an item set in the transactional database and choose the minimum support and confidence level;
2. Take all supports in the transaction that have a greater support value than the minimum or specified support value;
3. Discover all of the rules in these subsets with a greater confidence value than the specified threshold;
4. Arrange the rules in decreasing order of lift.

In this research work, the Apriori algorithm is applied in the R language to obtain frequent patterns of criminal activities based on the factors mentioned in Table 3. To verify the frequent patterns mined, support, lift, and confidence values of each rule are computed for the association rule $x \rightarrow y$.

For rule $x \rightarrow y$, the support of the rule is indicated as $\sup(x \rightarrow y)$. The support value shows the frequent occurrence of x and y in the dataset together. According to Equation (1), the relative support is defined as the transaction frequency $X \cup Y$ that appears in the database divided by the total number of transactions [22].

$$Support = \frac{(X \cup Y)}{N} \tag{1}$$

Based on the support-based downward closure property of itemsets, the Apriori method constructs frequent itemsets in a reasonable time. An item-set is frequent if its support value is equivalent to or above the pre-defined minimum support. An item-set Z of length l is frequent if and only if all the subsets of Z with length l-1 are frequent. This property allows the search space to be pruned substantially. Confidence indicates the

accuracy of the rule. The rules are ranked based on their confidence value. Confidence is computed using the Equation (2) formula [22].

$$Confidence(x \rightarrow y) = \frac{XUY}{X} \tag{2}$$

Thus, confidence is described as the number of transactions where XUY appears divided by the number of transactions where X appears. If multiple rules have identical confidence values then they are arranged based on their support value. To find the interestingness of an association rule generated, a lift measure is used. Lift is computed based on support values computed collectively as well as individually of each frequent item in the item-set as shown in Equation (3) [22].

$$Lift = \frac{Support(X \ U \ Y)}{Support(X) * Support(Y)} \tag{3}$$

Support (*XUY*) is the relative count of transactions, Support(X) is the number of transactions containing X, and Support (Y) is the number of transactions containing Y. The greater the value of the lift, the higher the interestingness of that rule. The Apriori algorithm is applied to different levels of Crime Type to get the desired results as shown in Figure 2.

With the help of association rule mining, risk factors related to crime are identified by analysing the crime patterns and possible crimes that happened at different times and days. For example, if 'Theft' *Crime Type* had happened at *Crime Time* 'Evening' and *Day Type*'Weekend', then other types of criminal activities can be expected to happen on the same day and time. The patterns obtained from the Apriori algorithm are considered for the identification of possible types of crimes (*Crime Type*) that happened on a particular time (*Crime Time*) and day (*Day Type*) based on the attributes *Crime Time*, *Day Type*, *Distance of Crime Event from Police Station* and *Direction of Crime Event from Police Station*.

Frequent patterns of length two to length five are generated by taking into consideration the crime dataset. The results of the Apriori algorithm are presented in the Section 4.

## 4. Results and Discussion

In this section, the results obtained from the cluster analysis and association rule mining are discussed in detail. This analysis has helped in identifying the risk of potential criminal activities as well as frequent crime patterns of different crime types by taking into account the factors mentioned in Table 3.

### 4.1. Cluster Analysis Result

To obtain the number of clusters (k), we have applied the elbow method in Python to find the best K value. The cost is plotted for a range of K values as shown in Figure 12. The cost is equal to the total of all the dissimilarities within the cluster method. The elbow method plots the cost function value generated by various k values. The number of cluster (k) values is said to be the most acceptable when the cluster value has the greatest decline and forms an angle [34].

In the above graph, we can see a bend at k = 5 indicating the optimal number of clusters. Therefore, an appropriate value to pick here for cluster analysis is k = 5.

Table 4 shows the results of the k-modes algorithm applied to the crime dataset for k = 5. From the results of the clustering analysis can be seen that *Crime Type* 'Theft' is majorly occurring in the different *Crime Event Location* 'Multan Chungi' and 'G1 Market'. Likewise, *Crime Type* 'Robbery' is mostly occurring in the different *Crime Event Location* 'Mughalpura Road' and 'Abbas Block'. Hence it can be concluded that theft and robbery criminal activity is majorly happening in 'Mustafa Town', 'Mughal Pura', and 'Johar Town' areas. Therefore, in case of theft and robbery criminal activity, these particular areas are identified as risky locations.

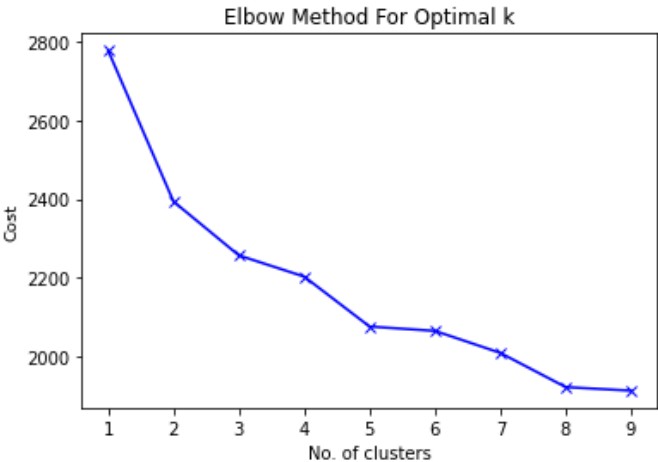

**Figure 12.** Obtained number of clusters (k) using the elbow method.

**Table 4.** K-modes clustering results of crime events for k = 5.

| Attribute | Cluster0 (138) | Cluster1 (207) | Cluster2 (130) | Cluster3 (78) | Cluster4 (83) |
|---|---|---|---|---|---|
| *Town* | Mughal Pura | Mustafa Town | Mustafa Town | Johar Town | Mustafa Town |
| *Crime Time* | Night | After Noon | Morning | Morning | Night |
| *Day Type* | Weekend | Weekdays | Weekdays | Weekdays | Weekend |
| *Crime Event Location* | Mughalpura Road | Multan Chungi | Wahdat Road | G1 Market | Abbas Block |
| *Distance of Crime Event from Police Station* | 2 km | 1 km | 3 km | 1 km | 3 km |
| *Direction of Crime Event from Police Station* | EAST | WEST | NORTH | EAST | SOUTH |
| *Crime Type* | Robbery | Theft | Fraud | Theft | Robbery |

Based on the results obtained from the k-modes algorithm shown, an analysis has been made for the identification of possible criminal activities that can also happen at a particular location. Against each clustering solution, we have extracted unique groups of related records with the help of obtained K-modes clustering results in a CSV file generated in Google Colab.

For the identification of the criminal activities (*Crime Type*) that can also happen in a particular location (*Crime Event Location*), we have fixed the values of the attributes *Crime Time, Day Type, Distance of Crime Event from Police Station,* and *Direction of Crime Event from Police Station* as shown in Table 4. The subsection below discusses the risk identification of possible criminal activities inparticular locations from the clustering solution in detail.

### 4.1.1. Risk Identification of Possible Criminal Activities from Clustering Solution

The clustering approach is used to identify crime risks concerning different crime types, locations, and times. Suppose that we have two locations (e.g., A and B) in cluster 1 based on their similarities in attribute values. Let crime types that occurred at location A be {C1, C2, C3, C4} and crime types that occurred at location B be {C1, C2, C3, C5}. So, based on the similarity of {C1, C2, C3} values at both locations, C5 can be predicted for location A as a potential risk and C4 can be identified as a potential risk for location B. However, this would be a weak assumption that can further be validated using the association analysis which is presented in Section 4.2.

From the clustering solutions shown in Table 4, we have extracted a list of unique groups of related records from each cluster with the help of K-modes clustering algorithm results obtained in a CSV file.

Table 4 shows some sample records from the cluster which helped in the risk identification of possible criminal activities that can also happen at particular locations.

The Cluster 0 records present in Table 5 are linked based on the values of the attributes *Crime Time*, *Crime Day*, *Distance of Crime Event from Police Station*, and *Direction of Crime Event from Police Station*. Keeping in account this established link, the risk of occurrence of criminal activities (mentioned in the last column) is identified. For example, in record 1, it has been identified that along with 'Robbery' crime, 'Gambling', 'Fraud', and 'Traffic Violation',are also expected to happen in the 'Multan Chungi' location based on the values of the attributes *Crime Time* = 'After-Noon', *Crime Day* = 'Weekdays', *Distance of Crime Event from Police Station* = '3 km', and *Direction of Crime Event from Police Station* = 'SOUTH'. The frequency of occurrence of crime Type 'Gambling', 'Fraud', and 'Traffic Violation' is high for the *Crime Time* = 'After-Noon', *Crime Day* = 'Weekdays', *Distance of Crime Event from Police Station* = '3 km', and *Direction of Crime Event from Police Station* = 'SOUTH'.

**Table 5.** Sample list of identified criminal activities that can also happen inparticular locations from cluster 0.

| # | Crime Event Location | Crime Time | Day Type | Distance of Crime Event from Police Station | The direction of Crime Event from Police Station | Crime Type | Risk of Possible Criminal Activities |
|---|---|---|---|---|---|---|---|
| 1 | Multan Chungi | After-Noon | Weekdays | 3 km | SOUTH | Robbery | Gambling, Fraud, Traffic Violation |
| 2 | Wahdat Road | Morning | Weekdays | 2km | NORTH | Kidnapping | Robbery, Possession of Weapon |
| 3 | BOR Main Road | Morning | Weekend | 1 km | EAST | Fraud | Drug Traffic king, Kidnapping, Theft |
| 4 | G1 Market | Night | Weekdays | 1 km | WEST | Theft | Possession of Weapon, Kidnapping |
| 5 | Ramgarh Bazar | Morning | Weekdays | 1 km | EAST | Traffic Violation | Fraud, Price Control, Theft |
| 6 | Mughalpura Chowk | Night | Weekend | 1 km | WEST | Gambling | Assaults, Drug Trafficking, Theft |

*4.2. Apriori Algorithm Results*

After performing the cluster analysis, association rule mining is applied to the crime dataset for the generation of frequent crime patterns. The objective of the generation of frequent patterns is to analyse the criminal activity patterns which are mostly occurring at a particular time (*Crime Time*) and day (*Day Type*). Frequent pattern analysis has also helped in the verification of the results obtained from the clustering solutions shown in Tables 4 and 5, respectively.

The Apriori algorithm is applied to different levels of *Crime Type* (shown inFigure 3) for the frequent pattern generation of criminal activities. For pattern identification of different crime types, the attribute *Crime Type* is fixed on the right-hand side of the association rules while the rest of the attributes shown in Table 3 are set as default on the left-hand side. Table 6 shows a sample list of strong and interesting crime patterns generated using the Apriori algorithm in R language along with the support (S), confidence (C), and lift (L) values computed. To find the frequent item sets in the crime dataset, the minimum support threshold is set to 3.

Based on the sample frequent patterns shown inTable 6, the patterns of criminal activities are identified. For example, rule #6 is showing that if *Crime Time* is 'After-Noon', *Day Type* is 'Weekdays', *Direction of Crime Event from Police Station* is 'SOUTH, WEST' and *Distance of Crime Event from Police Station* is '3 km', then there is a high probability of occurring of 'Fraud' (*Crime Type*) based on the support 7, confidence 1 and lift 59.1, respectively.

*4.3. Validation of Results*

The results obtained through the association rule mining validate the risky locations identified through the cluster analysis. For cluster analysis, in the example discussed in Section 4.1.1 of record 1, it has been identified that *Crime Type* 'Fraud' is expected to happen on the 'Multan Chungi' location based on the values of the attributes

*Crime Time* = 'After-Noon', *Crime Day* = 'Weekdays', *Distance of Crime Event from Police Station* = '3 km', and *Direction of Crime Event from Police Station* = 'SOUTH'.

**Table 6.** Sample of strong and interesting association rules mined from the crime dataset.

| # | Strong Association Rules (in Descending Order of Lift Values) | S | C | L |
|---|---|---|---|---|
| 1 | "{DayType = Weekend, Distance = 3km} => {CrimeType = Kite Flying offense}" | 9 | 1 | 75.429 |
| 2 | "{DayTime = Morning, DayType = Weekend, Direction = south} => {CrimeType = Marriage ordinance}" | 9 | 1 | 69.123 |
| 3 | "{DayType = Weekdays, Direction = SOUTH EAST, Distance = 3km} => {CrimeType = Conspiracy}" | 8 | 1 | 68.67 |
| 4 | "{DayTime = Morning, DayType = Weekend, Direction = North, Distance = 2km} => {CrimeType = Kidnapping}" | 8 | 1 | 67.12 |
| 5 | "{DayTime = After-noon, DayType = Weekdays, Direction = EAST, Distance = 1km} => {CrimeType = Robbery}" | 8 | 1 | 60.28 |
| 6 | "{DayTime = After-Noon, DayType = Weekdays, Direction = SOUTH, Distance = 3 km} => {CrimeType = Fraud}" | 7 | 1 | 59.1 |
| 7 | "{DayTime = Night, DayType = Weekend, Direction = WEST, Distance = 1km} => {CrimeType = Theft}" | 7 | 1 | 53.12 |

Similarly, the Apriori algorithm result discussed in Section 4.2, according to rule #6, shows that if *Crime Time* is 'After-Noon', *Day Type* is 'Weekdays', *Direction of Crime Event from Police Station* is 'SOUTH, WEST' and *Distance of Crime Event from Police Station* is '3 km', then there is a high probability of occurring of 'Fraud' (*Crime Type*). Hence, the same result of the occurrence of *Crime Type*, 'Fraud' has been identified through the cluster analysis as well as association rule mining. Cluster analysis is performed to identify the risk of occurrence of possible criminal activities in a particular location while association rule mining gives frequent patterns of criminal activities with the strength of the association between two events. The results yielded from both analyses support each other which implies that the identified similar criminal activities and crime patterns are expected to happen.

## 5. Limitations

Due to data privacy concerns, the availability of crime data was limited. Thus, data collection was a very challenging task for this research. Apart from this, existing crime records were non-digital and were not present in the English language. Therefore, a huge amount of effort was required for the English translation and conversion to digital format.

## 6. Conclusions and Future Work

Data mining techniques and tools have brought remarkable changes to the way data are analysed as well as revealing useful information. The main aim of this study is to provide a useful analysis that can lead to the development of efficient solutions to help concerned police teams and analysts solve crime cases by analysing similar criminal activities and crime patterns. For this matter, crime data from different police stations in Lahore, Pakistan, are collected and thoroughly analysed. Primarily, the data were available in the hard format written in the Urdu language. Data pre-processing was applied to the gathered crime data for its conversion into structured tabular form. To extract trends present in the dataset, data visualization analysis was performed which helped in effective exploratory analysis. Data mining techniques, namely cluster analysis and association rule mining, were applied for automated risk identification of criminal activities that are expected to happen at a particular location, time, and day. K-modes clustering algorithm is applied to group similar criminal activities. Apriori algorithm is used for the extraction of the frequent crime patterns and associations among crime type, day type, time of day, and crime location. This work can be an important step forward to reduce the growing rate

of crime in the world. The output of this research is very helpful for the concerned police officers and analysts in cracking and solving crime cases with less effort.

In the future, the dimensionality of the dataset needs expansion for analysing the crime patterns involved in different cities of Pakistan based on hotspots. The deep learning approaches could be developed and identify risky locations of a real-time crime at particular region from an image. With the help of solved crime case records, the crime pattern of a particular criminal can also be identified which leads to the identification of criminal network groups. Furthermore, machine learning techniques like classification and regression analysis can be applied to get useful trends from the data.

**Author Contributions:** Author Contributions: Conceptualization, F.F., Methodology, M.T.H., Software, S.M., Validation, H.A., Formal Analysis, M.I., Investigation, Resources, M.A., Data curation, A.M., Writ-ing-Original, M.T.H., Original Draft Preparation, F.F., Review and Editing, M.I., Supervision, E.-A.A. All authors have read and agreed to the published version of manuscript.

**Funding:** This research received no external funding.

**Institutional Review Board Statement:** Not Applicable.

**Informed Consent Statement:** Not Applicable.

**Data Availability Statement:** Not Applicable.

**Conflicts of Interest:** The authors have no conflict of interest.

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
