# Peer review of "Risk and Pattern Analysis of Pakistani Crime Data Using Unsupervised Learning Techniques"

_applsci, doi:10.3390/app12073675_

Round 1

Reviewer 1 Report

This paper attempts to analyze the risk and pattern of criminal activities using the K-modes clustering algorithm and Association Rule Mining technique. Although this study has potential contribute to the literature, there are still some deficiencies.

(1) There are many instances of language errors in this article. For example, the two words are wrongly connected. The grammatical problems must be avoided.

(2) The structure of the article is not appropriate. For example, too much description of crime data and background in the introduction section, but lack of description of research innovation. In literature review section, the article just lists many existing studies, lacks a summary of the current research status, and the review is not comprehensive enough. The method section contains too much description of visualization results.

(3) Many figures in the article lack clear information in vertical axis, such as figure 4 and figure 5. Moreover, the font size of the labels in some figures is chaotic. The labels are covered up and cannot be seen clearly, such as figure 8 and figure 10.

(4) The number of some types of crime is small and the occurrence of crime may be accidental, so the frequent patterns of criminal activities obtained by Association Rule Mining may be unreliable. The significance testing is further needed.

Author Response

Response to Reviewers

Original Manuscript ID:applsci-1631508

Original Article Title: “Risk and Pattern Analysis of Pakistani Crime Data Using Unsupervised Learning Techniques”

Dear Editor,

We would like to thank you for your valuable feedback on our manuscript submitted to the Applied SciencesJournal. We have noted the comments from the reviewers and subsequently addressed all the highlighted issues. In addition, we have endeavored to correct parts that were unclear and have added necessary information as suggested by the reviewers. Please find enclosed the revised draft and our response to the comments.

We appreciate the reviewers for careful review of the manuscript and providing constructive feedback.

Best regards,

Faria Ferooz

Muhammad Idrees

Reviewer 2 Report

  1. Please do another round of proofreading. I highlighted some typos, confusing sentences that need to be rephrased. I highlighted some of them. 
  2. The feature selection is consulted with experts. No machine learning methods? What is innovation in here? 
  3. The validation process is not convincing. They should split the data into training and testing, then use the algorithm to predict the testing and check out the accuracy.

Author Response

(The authors gave the same response as above.)

Round 2

Reviewer 2 Report

  1. Fix the format issue on line 105 and line 109: “Error! Reference source not 105 found..”
  2. There are either format issues or some typo issues. There is no space between words. For example:  line 132 “random forestare” maybe “forest are” line 189 “determineextracted”, line 214 ((high, and line 323 “anddetermine” line 334 “happenin”  and line 342 “expertsto” line 396, “Level 3respectively” and “therisky” and line 585 “criminalactivities” and line 587  “ofrelated”, etc. 

Author Response

Original Manuscript ID:applsci-1631508

Original Article Title: “Risk and Pattern Analysis of Pakistani Crime Data Using Unsupervised Learning Techniques”

Dear Editor,

We would like to thank you for your valuable feedback on our manuscript submitted to the Applied Sciences Journal. We have noted the comments from the reviewers and subsequently addressed all the highlighted issues. In addition, we have endeavored to correct parts that were unclear and have added necessary information as suggested by the reviewers. Please find enclosed the revised draft and our response to the comments.

We appreciate the reviewers for careful review of the manuscript and providing constructive feedback.

Best regards,

Muhammad Idrees
